# High Levels of Anabaenopeptins Detected in a Cyanobacteria Bloom from N.E. Spanish Sau-Susqueda-El Pasteral Reservoirs System by LC–HRMS

**DOI:** 10.3390/toxins12090541

**Published:** 2020-08-22

**Authors:** Cintia Flores, Josep Caixach

**Affiliations:** Mass Spectrometry Laboratory-Organic Pollutants, IDAEA-CSIC, Jordi Girona 18, 08034 Barcelona, Spain; josep.caixach@idaea.csic.es

**Keywords:** cyanobacteria, bloom, microcystins, anabaenopeptins, oscillamide Y, cyanobacterial peptides, LC–HRMS, Orbitrap

## Abstract

The appearance of a bloom of cyanobacteria in the Sau-Susqueda-El Pasteral system (River Ter, NE Spain) in the autumn of 2015 has been the most recent episode of extensive bloom detected in Catalonia. This system is devoted mainly to urban supply, regulation of the river, irrigation and production of hydroelectric energy. In fact, it is one of the main supply systems for the metropolitan area of cities such as Barcelona and Girona. An assessment and management plan was implemented in order to minimize the risk associated to cyanobacteria. The reservoir was confined and periodic sampling was carried out. Low and high toxicity was detected by cell bioassays with human cell lines. Additionally, analysis studies were performed by enzyme-linked immunosorbent assay (ELISA) and liquid chromatography–high resolution mass spectrometry (LC–HRMS). A microcystin target analysis and suspect screening of microcystins, nodularins, cylindrosperpmopsin and related cyanobacterial peptides by LC–HRMS were applied. The results for the analysis of microcystins were negative (<0.3 μg/L) in all the surface samples. Only traces of microcystin-LR, -RR and -dmRR were detected by LC–HRMS in a few ng/L from both fractions, aqueous and sestonic. In contrast, different anabaenopeptins and oscillamide Y at unusually high concentrations (µg-mg/L) were observed. To our knowledge, no previous studies have detected these bioactive peptides at such high levels. The reliable identification of these cyanobacterial peptides was achieved by HRMS. Although recently these peptides are detected frequently worldwide, these bioactive compounds have received little attention. Therefore, more studies on these substances are recommended, especially on their toxicity, health risk and presence in water resources.

## 1. Introduction

Cyanobacteria, initially known as blue-green algae, are unicellular oxygenic prokaryotic organisms considered to be important contributors to the formation of the earth’s atmosphere, since they are the first known organisms capable of photosynthesis [1], and of nitrogen fixation dating back 3.5 trillion years in evolution. Of the many genera of known cyanobacteria, *Microcystis* spp., *Planktothrix rubescens*, *Aphanizomenon* spp. and *Anabaena* spp. are quite common in Europe. Over the last few decades, cyanobacteria are under growing investigation due to massive occurrence in eutrophic waters worldwide and for climate change [2,3,4,5]. Under certain conditions of growth and environment factors, cyanobacteria can tend to proliferate massively (blooms). Some factors that may favor the formation of blooms are the increase in the concentration of nutrients (especially the phosphorus), a low relation nitrogen/phosphorus, the increase of temperature and conditions of hydrological stability (high residence times or stable stratification of the water body). The presence of these blooms is an important concern for water quality and has become a serious ecological, health and economical issue because cyanobacteria are able to produce bioactive substances associated with taste and odor algae problems and diversity of toxins [6,7,8] as secondary metabolites. Cyanotoxins are toxic to humans and animals [9]. They are commonly referred as hepatotoxins, neurotoxins, cytotoxic alkaloids (affecting different organs: liver, kidneys, adrenal glands and small intestine) and dermatotoxins [10]. For example, acute and chronic liver damage is associated with exposure to hepatotoxic peptides microcystins (MCs) and the alkaloid cylindrospermopsin [11,12,13,14]. MCs and cylindrospermopsin (CYN) are the most widely distributed algal toxins in freshwaters, and nodularins (NODs) are considered mainly marine/brackish toxins, but an increasing number of potentially toxic secondary metabolites has been reported (anabaenopeptins, microviridins, agardhipeptins, oscillapeptins, oscillapeptilides, oscillamide, oscillaginin and oscillacyclin, for example) [15,16]. Certainly, MCs rarely appear alone during cyanobacteria blooms, and other related cyclic or linear oligopeptides can even be dominant. Although these bioactive substances are not included in any regulations or recommendations and they are considered less toxic than MCs, which are a potent and regulated toxic, current data show that they can give a positive response in some toxicity tests [11]. Specifically, anabaenopeptins (APs) have been shown to be inhibitors of protein phosphatases and carboxypeptidase A [11,15,16]. However, these cyanopeptides have received little attention.

To protect consumers from the adverse effects of cyanobacterial peptide toxins, the World Health Organization (WHO, 1998) recommended a maximum guideline level in drinking water of 1 μg/L for total MC-LR (free plus cell-bound) [17]. Accordingly, several worldwide countries such as China, Korea, France, Japan, Norway, Poland, Czech Republic, New Zealand, Brazil and Spain have established guideline values in their national regulations for some toxins in public water bodies for human consumption and for recreational uses [18]. The regulated levels of cyanotoxin (anatoxin-a, CYN, MC-LR and saxitoxins) in the drinking water range from 1.0 to 15 µg/L. Specifically, Spain was one of the first European countries to adopt a parametric value for MC. The Spanish authorities established a maximum level for MC in drinking water of 1 µg/L (RD 140/2003) [19] and include three levels of probability for cyanobacterial proliferation in water-bodies used for recreation (RD 1341/2007) [20]. Paradoxically, until now the European regulation did not include a guideline value for cyanotoxins. Recently, European Commission adopted on 1 February 2018 a proposal for a revised drinking water directive to improve the quality of drinking water [21]. The proposal document includes a maximum residue level of 1.0 μg/L for MC-LR.

For cyanotoxins detection the most common methods include an enzyme-linked immunosorbent assay (ELISA), protein phosphatase inhibition assay and liquid chromatography–mass spectrometry (LC–MS) [22]. ELISA assay is a colorimetric method that uses specific antibodies to detect the presence of characteristic amino acid Adda of MCs. ELISA tests are widely employed as a screening tool for the analysis of MCs [23]. This bioassay is a sensitive, easy-to-use and inexpensive method. Unfortunately, it is unselective toward specific MC variants so the result is usually expressed as MC-LR equivalents. Accordingly, LC–MS is a selective and specific methodology that can provide reliable detection, differentiation and quantitation of all cyanobacterial secondary metabolites [18,24]. Tandem mass spectrometry (MS/MS) has been widely adopted for the routine quantitative analyses of target cyanotoxins in water samples using the very sensitive multiple reaction monitoring (MRM) mode of triple quadrupole (QqQ) mass spectrometers [23,25,26,27,28,29,30,31]. For structural characterization and quantitation purposes, 3D ion trap (IT) [32,33,34,35,36,37], linear ion trap (LIT) [38], time-of-flight (TOF) [39,40], Q-TOF [41,42,43,44,45] and Orbitrap mass spectrometers [26,46,47,48,49,50,51,52] were employed. High mass accuracy of TOF and Orbitrap instruments is beneficial for LC–MS and LC–MS/MS analyses of cyanotoxins [24]. In fact, high resolution full scan strategies have strongly emerged for the screening non-targeted analysis and detection of not commonly monitored or known cyanotoxins. However, there is limited availability of accurate mass spectra databases and few studies apply liquid chromatography–high resolution mass spectrometry (LC–HRMS) analytical protocols for suspect screening of cyanotoxins and cyanopeptides in freshwaters [41,49].

Globally, the presence of cyanobacteria in the water bodies of many countries has been extensively studied [38,43,53,54,55,56,57,58,59]. At least eight countries have initiated national programs on water quality and cyanobacterial problems (Australia, Brazil, Canada, Finland, Great Britain, Japan, Portugal and the United States). For its part, Spanish reservoirs present a high probability of developing blooms of toxic cyanobacteria [60,61,62,63]. The studies about the abundant appearance of cyanobacteria in different Spanish Basins show that cyanotoxins are often present and occasionally show very high toxin concentrations. Although the Spanish water reservoirs are often affected by cyanobacteria blooms, there is little work published about cyanobacterial bloom episodes [64,65,66].

This study reports a description of an extensive and toxic bloom of cyanobacteria in the strategic system of reservoirs from Sau-Susqueda-El Pasteral (Catalonia) and the protocol actions and exhaustive monitoring implemented to assess and manage the risk related to water contaminated by cyanobacteria. In the present work, the important role of HRMS is shown for target analysis and suspect screening of cyanotoxins, including not only MCs but other characteristic cyanotoxins and cyanopeptides families (NODs, CYN, APs, oscillapeptins, microviridins, agardhipeptins, oscillapeptilides, oscillamide, oscillaginin and oscillacyclin) in a real case study. A non-target methodology able to perform a retrospective analysis of any toxin and other pollutants was applied. The accurate mass spectra data were obtained using an Orbitrap mass spectrometer (maximum resolving power of 100,000 and ±5 ppm extraction window). Unexpectedly, only traces of MC-LR, -RR and -dmRR were detected. In contrast, unusually high intracellular concentrations (µg-mg/L) of APs (A, B, C and F) and oscillamide (OCA) Y were confirmed. To our knowledge, such high concentrations of cyanobacterial peptides (CPs) have never been reported before. Finally, it is important to highlight that effective consumer protection requires efficient detection of the whole spectrum of cyanobacterial toxin congeners, especially, considering that different bioactive metabolites are often detected simultaneously [31,32,35,38,41,42,51]. The HRMS approach has been demonstrated to be particularly suitable for the detection and structural confirmation of other CPs, more than MCs, which are not commonly monitored.

## 2. Results and Discussion

The appearance of a massive growth of cyanobacteria in the Sau-Susqueda-El Pasteral system in autumn 2015 has been the most recent episode of bloom in Catalonia in terms of its extent and duration that affected such a strategic water body. As already mentioned before, this system of reservoirs is mainly devoted to the urban supply, but also to regulation of the River Ter, irrigation and production of hydroelectric energy. In early September 2015, a massive proliferation of cyanobacteria was detected in the Susqueda reservoir, which disappeared within a few weeks. Later, in October, the abnormal presence of algae was detected in the Sau reservoir in a more significant duration and magnitude. Both the coloration and the characteristic smell that they gave rise to think of an episode of bloom. According to other authors, the bloom appeared in the months of maximum risk for cyanotoxin occurrence when *Microcystis* typically dominated the phytoplankton community [61]. The microscopic analysis confirmed that the bloom corresponded to colonies of the genus *Microcystis*, most likely *M. aeruginosa* or *M. flos-aquae*, potentially cyanobacteria-producing MC in an active phase of growth. In the month of November this situation persisted.

LC–HRMS with high mass accuracy has emerged as a powerful tool for environmental analysis. Highly resolved and accurate Orbitrap technology used allows for a more reliable target analysis with reference standards, a screening for suspected analytes without reference standards, or a screening for unknowns.

### 2.1. Microcystin Target Analysis by LC–HRMS

The MCs more commonly detected in blooms and those that we have a certified analytical standard were analyzed by the target LC–HRMS strategy. The list of target MCs were presented in Table 1 and Table 2.

As already mentioned above, in early September, the beginning of a bloom was detected in the Susqueda reservoir. According to RD 140/2003 that establishes the health criteria of the quality of water for human consumption in Spain [19], as this system of reservoirs is devoted to urban supply, when there is eutrophication in the catchment water, the level of MC concentration must be controlled. As already described in the introduction, MCs are included in the RD at a maximum level of 1 µg/L. In this context, three samples were received from the water catchment of the El Pasteral reservoir and from the dam of the Susqueda reservoir (samples of the El Pasteral reservoir: Input Drinking Water Treatment Plant and Costa Brava tank; and samples of the Susqueda reservoir: Dam). As described in Section 4.3 the samples were filtered for the independent analysis of aqueous (extracellular) and sestonic (intracellular) fractions. Table 1 includes the results of target MC analysis by LC–HRMS for aqueous fractions. As shown in Table 1, no MCs were detected in the extracellular fraction of these three samples (limit of quantitation, LOQ: 0.3 µg/L), thus complying with the legislated value. On the other hand, the presence of target MCs is also not observed in sestonic samples (LOQ: 10 ng/L). It should be noted that the limit of quantitation (LOQ) of methodologies for analysis of extracellular (direct injection and LC–HRMS analysis) and intracellular (preconcentration by methanol extraction and LC–HRMS analysis) fractions were different, 10 ng/L and 0.3 µg/L, respectively. A preconcentration was applied in the analysis of the sestonic fraction. Therefore, a lower limit of quantitation for this methodology could be achieved.

Regarding the toxicity from Sau bloom detected later in October, ELISA tests performed were negative (<0.3 μg/L of MCs) in all depths, but cell bioassays performed with human cell lines detected some toxicity, both in the extracellular fractions’ analysis (low toxicity) and in the intracellular fractions’ contents (high toxicity). As already described in the introduction, ELISA is a fast, simple and economically affordable analysis, but with a non-specific response and susceptible to matrix effects. It can respond to other toxins (false positives) and not all MC variants have the same analytical response (false negatives). Therefore, different representative locations in the Sau reservoir (maximum concentration, areas and depths by catchment supply) were sampled for the analysis of toxins by LC–HRMS. LC–HRMS analysis is a very sensitive, specific and selective methodology [24]. As shown in Table 1, target MC analysis by LC–HRMS were negative (<0.3 μg/L) in aqueous fraction of samples from the water supply catchment (A1), superficial scums (M1), reservoir center (M2) and at 11 m of depth (M3). Only traces of MC-RR (all samples) and -LR (sample M3) were detected in concentrations estimated to be close to 10 ng/L. Remember that MC are legislated by Spanish authorities in RD 140/2003 at a maximum level for drinking water of 1 µg/L [19]. MC-LR, the most widely reported MC variant in the literature and the most toxic, was not detected in the superficial samples and just observed in the only sample taken at a depth of 11 m (M3).

On the other hand, Table 2 summarizes the results of target analysis of MCs by LC–HRMS for the sestonic fraction of Sau samples. No filtrate was obtained when filtering sample A1, the water had neither algae nor particulate. Therefore, the intracellular fraction of this sample was not analyzed. Samples M1 and M2 were very intense green color and high algae content. In contrast, target analysis by LC–HRMS did not detect the presence of any of the most common MCs (see Table 2). Finally, in the M3 sample, with little particulate–algae, MC-RR and -dmRR were detected at very low concentrations (<LOQ: 10 ng/L, at 0.4 ng/L estimated concentration by extrapolation).

Due to very low concentrations of target MCs found, despite that a significant presence of *Microcystis* and toxicity was detected, it could be that the strains present were poorly toxin producing or there are only some colonies (few) that are toxic.

### 2.2. Suspect Screening by LC–HRMS

Additionally to target MC analysis, due to the great potential and versatility of LC–HRMS, a suspect screening of others toxins and CPs not considered in most monitoring plans was performed. The samples analyzed by LC–HRMS were acquired in the full scan mode, consequently it is possible to carry out a screening for the analysis of any ionizable substance, simply extracting its exact mass. In contrast to the tandem mass spectrometry analysis (MS/MS), which uses specific MS/MS conditions, the analysis in the full scan by HRMS uses general electrospray (ESI) conditions and voltages in all the *m*/*z* range, so there is no imperative to have standards of all analytes for its optimization. Although a large number of MCs variants (more than 200) and CPs have already been described in the literature, only a few analytical standards are currently available, less than twenty for the MC variants, in particular. However, in these environmental fields where reference standards are currently not available, compound-specific information for suspects is available, such as molecular formula and structure, which can be efficiently used in the identification and confirmation process. Suspect screening is done based on a suspect ion list or database that summarizes all this information from the literature and from in silico approaches. As mentioned later in Section 4.5, a homemade database of 157 MCs, 10 NODs, CYN and 29 CPs (10 APs, 8 oscillapeptins, 4 microviridins, 2 agardhipeptins, 2 oscillapeptilides, OCA Y, oscillaginin and oscillacyclin) was used for identification purposes in suspect screening. Even, the HRMS allows the non-targeted analysis to discover new and unknown toxins or a retrospective analysis afterwards for the analysis of any other compound that was not initially included in the study. LC–HRMS operates at high resolution and the full scan mode with high sensitivity, which allows their capabilities in the detection of unknowns. With regard to environmental monitoring programs, a major advantage of LC–HRMS is the possibility of a retrospective analysis of full-scan data, which enables laboratories to search for “new” contaminants years after data recording, without having to reanalyze samples. In none of the samples received for analysis by LC–HRMS were any of these 157 MC variants detected, nor were CYN, but related CPs were identified in almost of all samples at elevated concentrations in some intracellular fractions (until mg/L). These CPs are not commonly monitored, so they would not have been detected using a target analysis method such as MS/MS.

In the samples received in September from the Susqueda and El Pasteral reservoirs, the presence of APs at the trace level in the intracellular fraction was earlier detected. Additionally, in many other previous studies, these peptides have already been found in cyanobacterial blooms and it is typical to find them associated with MCs [31,35,41,42,51,65,67,68]. These peptides are bioactive and are commonly produced by species such as *Anabaena* and *Planktothrix* [11,15,16].

As for the Sau reservoir, the samples presented APs and OCA Y in both fractions, extra- and intracellular. For extracellular fractions, APs A, B and F were observed in M1 and M2 samples and only AP F was detected in M3 and A1 samples. The level of AP F in M3 and A1 extracellular fractions was lower than in the two other Sau samples (M1 and M2). In fact, M1 and M2, superficial scums and a sample of reservoir center, respectively, were the more concentrated samples. On the other hand, the presence of high concentrations of several APs (A, B, C and F) and OCA Y were estimated in intracellular fractions of M1, M2 and M3 samples (Table 3). Note that high concentrations of this CPs were observed, especially in the superficial scums (M1) and in the center of the reservoir (M2) samples, from a few µg/L of AP C in M2 to 58 mg/L of AP F in the M1 intracellular fractions. As expected, due to the high content of algae presented, the superficial scums sample is the one with the highest concentration of bioactive metabolites at the intracellular level. Although recently these peptides are detected frequently worldwide [31,35,41,42,51,65,67,68], to our knowledge, no previous studies have detected these bioactive peptides at such high levels. In general, the main CPs detected for the samples analyzed were: AP F > AP B > OCA Y.

The reliable identification of these CPs was achieved by HRMS following the criteria described in Section 4.5. As an example, Figure 1 shows the extracted ion chromatogram (XIC) and mass spectra of CPs detected by LC–HRMS in M1 (superficial scums) sample. The extracted ion chromatogram signal of analyte (*m*/*z*) in the full scan acquisition mode was used for quantitation purposes. It should be highlighted that the identification in all samples was performed with excellent accuracy in the exact mass measurements (<2.4 ppm) of analyte and fragments *m*/*z* signals (see Figure 1, Figure 2 and Figure 3) and experimental isotopic pattern (IP) matching very well with the theoretical one (%IP > 95). Furthermore, elements in use, ring plus double bond equivalents (RDBE) and nitrogen rule were corresponded to analytes identified. Finally, their experimental fragmentation spectra agree with theoretical and experimental ones obtained by other authors [41,43,47]. The mass spectrum of each compound is their fingerprint and the different diagnostic fragments confirm their chemical structure. Characteristic fragments of APs (*m*/*z* 263.1390 and 201.0982) and OCA Y (*m*/*z* 651.3865 and 263.1390) were checked in their experimental all ion fragmentation (AIF) spectra. The strategy is to use diagnostic fragments to characterize the family of compounds that are detected (MCs, NODs, CYN and CPs) according to the homemade database. These fragments are usually common to the different congeners of the same family. Then, as the LC–HRMS method operates in the full scan acquisition mode at two collision energies, all the fragment ions were detected in the two product ion spectra to be able to identify the specific congener. For it, in the optimization of LC–HRMS conditions, the two collision energies that provided the most characteristic spectra were chosen. Alternating full scan and AIF acquisition modes allow quick and easy detection of signals that can be identified as cyanotoxins or/and CPs, extracting from the AIF chromatogram the exact mass (*m*/*z*) of its specific fragments. As Figure 2 shows, the main fragments of AP F at 30 and 70 eV were *m*/*z* 201.0982 ([CO-Arg]^+^, C7H13N4O3) and 70.0620 ([Arg chain]^+^, C4H8N). For OCA Y (see Figure 3), the main fragments at 30 and 70 eV were *m*/*z* 651.3859 ([Lys-Phe-MeAla-HTyr-allo-Ile]^+^, C35H51N6O6) and 84.0806 ([Lys chain]^+^, C5H10N). The characteristic fragment of *m*/*z* 263.1390 ([MeAla-HTyr]^+^) of APs and OCAs detected by other authors by collision-induced dissociation (CID) in the quadrupole collision cell [41] are a minority signal in the AIF acquisition mode using the Orbitrap analyzer at medium and high collision energies, 30 and 70 eV, respectively. The fragmentation by the Exactive mass spectrometer occurs in a special collision cell by high collision dissociation (HCD) without precursor mass selection (AIF mode).

The present results agree with recent studies in which analysis of cyanotoxins and CPs in surface waters reveals more than MC [31,32,35,41,42,51,65,67,68]. However, these cyanopeptides have received little attention and there are few toxicity and health risk studies about APs and OCA Y. Although these compounds are considered less toxic than MCs, recent data show that they may give a positive response in some toxicity tests such as phosphatase inhibition tests [11]. Additionally, just like MCs that are included in international recommendations and regulations, in contrast, to our knowledge, the APs are not regulated, although they can cause some toxicity.

Finally, in December when the algae almost disappeared, some samples were collected. LC–HRMS and ELISA analysis indicated the absence of MCs in all samples (LOQ: 0.3 µg/L). The presence of APs and OCA Y by LC–HRMS was also not observed. Therefore, as the analytics confirm the absence of toxins and CPs, and the water quality already allows its use without any risk associated with this phenomenon, the cyanobacteria bloom episode was considered the end and the system of reservoirs from Sau-Susqueda-El Pasteral happens to be managed normally.

### 2.3. Case Study Description

Initially, the algae populated a restricted and localized area of the reservoir (at one end of the dam). Days later, a substantial increase of bloom was occurred. The algae were widespread and big colonies with large numbers of cells invaded almost the entire surface layer of the reservoir up to 15 m of depth, and were especially prominent in the greater surface and on the shores and corners, where the wind accumulated them by making a surface crust of different thickness. During the bloom episode, an important reduction in nitrate entry load, high temperatures, low rate of water renovation and greater stability in the reservoir were observed. As is known, cyanophytes have the ability to capture nitrogen from the atmospheric origin, so the reduction between available nitrogen and phosphorus may have given them a clear advantage over other algal groups [1,8]. The water temperature in this reservoir system varies considerably between summer and winter. The maximum summer temperature was of the order of 5 degrees higher than the previous two years, about 21 °C peak in the summers of 2013 and 2014, and this summer of 2015 the entrance water to the reservoir reached 26 °C. It should be noted that the temperature data recorded on the surface of the Sau reservoir during 2015 have been the highest on record since data were available. Late autumn, the rains cause that the thermocline to break down and the inflow of the river, with a significant increase in turbidity and a decrease in its conductivity, leads to algae sinking. Finally, at the end of December, the temperature dropped considerably (3.5 °C in the morning), the water reservoir is completely mixed, the algae has substantially reduced their concentration throughout the water column, likely sank or been diluted, and toxicity and cyanopeptides were not detected. Therefore, the analysis confirms the end of bloom episode and the quality of the water already allowed its use without any risk associated with algae.

Historically, in the years 1998 and 2000, two episodes of massive growths of cyanobacteria were already observed, with the detection of MC-LR and -RR dissolved in water at levels ranging between 0.189 and 270 ng/L [69]. From then on, the improvement of sanitation in the upper Ter River basin and the progressive reduction of the nutrient supply had caused a reduction in the frequency and scope of these blooms, to the point that they had not been detected for more than a decade, at least with an important extension.

Several factors could be the cause of the massive proliferation of algae detected in the Sau-Susqueda-El Pasteral system. On the one hand, biotic factors as the predator/prey competition (e.g., phytoplankton/zooplankton) [70] may have contributed. However, these factors could also have accelerated the growth of other algal groups, which was not the case. On the other hand, considering that, as we have already mentioned, historically there have been previous episodes of bloom and some of the cyanobacteria detected in this reservoir system have mechanisms of resistance to the forecast of more suitable conditions [6], it was expected that changes in conditions (increase in nutrients or changes in temperature, etc.) would have enabled the algae formation. Therefore, it seems that the high temperatures, along with the reservoir stratification and stabilization throughout the summer and autumn 2015, could have conditioned the high cyanophyte bloom [71]. In fact, in the end of the bloom episode, in December, it was quite cold (3.5 °C in the morning).

### 2.4. Assessment and Management Tasks

During the 2015 Sau-Susqueda-El Pasteral bloom episode, the current action protocol on the risk associated to water contaminated by cyanobacteria was activated and the management and assessment plan was implemented in the reservoirs. The different actions were coordinated by a working group, composed by staff from Agència Catalana de l’Aigua, Aigües Ter Llobregat, laboratories and scientific experts. Several activities to protect the population were carried out with three main key goals: (a) prevent, as far as possible, the bloom spread throughout the reservoir system (Sau-Susqueda-El Pasteral), so as not to increase the bloom and that no algae downstream emerge; (b) keep the problem as far as possible from the catchment sites for water supply and (c) monitor the quality and potential toxicity of water for Sau-Susqueda-El Pasteral supplies and recreational activities.

Firstly, the reservoir was confined to prevent algae from spreading to other reservoirs, Susqueda and El Pasteral, and could reach the catchment areas of different supplies. Furthermore, gates were closed and the turbine was stopped.

On the other hand, given the potential for water toxicity, all water supplies were suspended. Therefore, all involved actors were informed: the municipalities and population users, CECAT (Centre superior de coordinació i informació de l’estructura de protecció civil de Catalunya), Departament de Salut de la Generalitat de Catalunya, the operators to the water supply downstream of the El Pasteral (Consorci de la Costa Brava and Aigües de Girona, Salt i Sarriá de Ter) and ENDESA (Spanish electricity company). They were alerted to the presence of a bloom algal that could cause water toxicity by ingestion, inhalation and contact, and were informed that the analyses were being performed to detect possible toxicity. The warning addressed potential uses for water and other activities in the reservoir. Consequently, the frequency of periodic sampling was significantly increased to almost weekly an exhaustive monitoring of the water was conducted.

Late autumn, with the arrival of rains and, as a consequence, the filling of the reservoir, management operations were performed: floodgates were opened and the turbinate was authorized to maintain the level of the reservoir and to prevent its overflow and the algae layer extends to Susqueda and El Pasteral reservoirs.

## 3. Conclusions

The bloom case study at the Sau-Susqueda-El Pasteral system during the autumn of 2015 could be considered to have been significant, both in size on surface (entire reservoir) and depth (15 m layer of depth), and for its duration, just over 3 months, which affected such a strategic water body. The implemented assessment plan and the intense monitoring made it possible to manage the reservoir while minimizing the risk of the cyanobacteria bloom episode and its downstream expansion.

During the bloom episode, large colonies with considerable numbers of cells invaded almost the entire surface layer of the reservoir up to 15 m of depth. Cell bioassays detected some toxicity, both in the extracellular content analysis (low toxicity) and in the intracellular content analysis (high toxicity). Despite the significant presence of *Microcystis* and toxicity detected, very low concentrations of MCs were found. The results for the analysis of MCs and CYN were negative (<0.3 μg/L) in all the samples. Only traces of MC-LR, -RR and -dmRR were detected by LC–HRMS in a few ng/L. Therefore, it could be that the strains present were poorly toxin producing or there are only some colonies (few) that are toxic. In contrast, different APs (A, B, C and F) and OCA Y at unusually high intracellular concentrations, in the order of mg/L were observed. The reliable identification of these cyanopeptides was achieved by HRMS. The applied target and suspect screening HRMS approach has been demonstrated to be particularly suitable for the detection of other cyanobacterial peptides, more than MCs, which are not commonly monitored by target strategies. The detection of the whole spectrum of cyanobacterial peptides is highly recommended for effective consumer protection. To our knowledge, no previous studies have detected these bioactive peptides at such high levels. Although these peptides are not included in any regulations or recommendations and they are considered less toxic than MCs, current data show that they can give a positive response in some toxicity tests. Therefore, more studies on these compounds are recommended, especially on their toxicity, health risk and presence in water resources.

Although no statistical evidences are available, high temperatures along with reservoir stratification and stabilization throughout the summer and autumn 2015 could be some of the factors that led to high cyanophyte bloom. In fact, it was the environmental conditions that finally led to the end of the episode: the winter temperatures dropped and the stratification was broken. This suggests that other episodes could impact this in the future, despite reducing the nutrient load on the reservoir, due to climate change and global warming. That is why it is advisable to take preventive actions that in the future will allow us to avoid this phenomenon as much as possible and to get ahead of the appearance again from bloom and palliative measures that, if appropriate, help to reduce the scope and duration of the episode and mitigate its consequences.

## 4. Materials and Methods

### 4.1. Study Site

The Ter River is 167 km long and originates in the middle of the Catalan Pyrenees, leading to the Mediterranean Sea. The system of reservoirs of Sau-Susqueda-El Pasteral is located in the Ter River basin (Figure 4) and is devoted mainly to the urban supply, regulation of the Ter River, irrigation and production of hydroelectric energy. In fact, it is one of the main supply systems for the metropolitan area of cities such as Barcelona and Girona. The whole population it is supplied to is about 5 million inhabitants.

Specifically, Sau and Susqueda, with a capacity of 165 and 233 Hm^3^ and a maximum surface area of 573 and 463 ha, respectively, are the major reservoirs of this system. As for the supply, the main function of reservoirs is basically indirect in Sau and Susqueda. Sau is the first of the three reservoirs. Therefore, it accumulates the first water that arrives from the Ter River, which will naturally begin to be self-purified, improving its characteristics for future catchment in the El Pasteral. It should be noted that the Ter River before the reservoirs system receives the direct impact of the metallurgic, pulp mill, textile and tannery industries. The Sau dam, over 60 m in height, it is provided with a spillway, bottom drains on both sides that can let the water pass to Susqueda directly, and a half-bottom drain at the same height as the previous ones but on the central side of the dam, which lets water through its valves, which allow oxygenation (Howell Bunger valves). In addition, it has a catchment tower, with gates at three different depths, through which the water, before passing to Susqueda, can be turbined for the generation of hydroelectric power.

### 4.2. Chemicals and Standards

All reagents employed were of analytical or high-performance liquid chromatographic (HPLC) grade. Acetonitrile and methanol were obtained from Merck (Darmstadt, Germany). Formic acid was from Panreac (Montcada i Reixac, Barcelona, Spain). High purity water produced with a Milli-Q Synergy UV system (Millipore, Bedford, MA, USA) was used. MC-LA, -LR, -RR, -WR, -YR and NOD standards were purchased from Alexis Biochemicals (San Diego, CA, USA). Stock standard solutions of each analyte (100 or 500 µg/mL) were individually prepared by weight in methanol and stored at −20 °C. Intermediate solutions were prepared weekly from the stock standard solution by appropriate dilution in methanol. MC-LF, -LW and -LY 5–10 µg/mL methanolic solutions were supplied from Sigma–Aldrich (St Louis, MO, USA). MC-dmRR 10 µg/mL in methanol was acquired from Cyano Biotech GmbH (Berlin, Germany). Finally, MC-dmLR 10 µg/mL methanolic solution was purchased from DHI (Hørsholm, Denmark). Mixed calibration standard solutions of all MCs were prepared daily. Nitrogen (purity > 99.999%) supplied by Air Liquide (Madrid, Spain) was used for the ESI source and as a CID gas in the Orbitrap mass spectrometer.

### 4.3. Extraction Procedure

All samples were collected in amber glass bottles, immediately placed on ice, and transported to the laboratory where they were filtered through Whatman GF/F filters (0.7 µm) for an independent analysis of aqueous (extracellular) and sestonic (intracellular, defined as the particles retained on 0.7 µm filters) fractions. The filter with the sestonic fraction was then stored at −20 °C and filtered water (extracellular fraction) was analyzed immediately by the LC–HRMS analysis via direct injection into the Orbitrap mass spectrometer. The filter was frozen–thawed three times to promote cell lysis, and then, the sestonic fraction was extracted three times with methanol acidified according to Barco et al. [72]. NOD was added as an internal standard in both, intra- and extracellular fractions for quantitation purposes. Toxins and CPs present in the extracts were identified and quantified by LC–HRMS.

### 4.4. LC–HRMS Analysis

An Orbitrap-Exactive HCD (Thermo Fisher Scientific, Bremen, Germany) mass spectrometer equipped with a heated electrospray source (H-ESI II), a Surveyor MS Plus pump and an Accela Open AS autosampler kept at 6 °C (Thermo Fisher Scientific, San Jose, CA, USA) were used for the LC–ESI–HRMS analysis [47]. The chromatographic separation was performed on a reversed-phase Phenomenex Luna C18(2) column (150 mm × 2.0 mm, 5 µm). The mobile phase was composed of Milli Q water as solvent A and acetonitrile as solvent B, both containing 0.1% (*v*/*v*) formic acid at a flow rate of 200 µL/min. The linear gradient elution program for the analysis was: 10–30% B 10 min, 30–35% B 20 min, 35–55% B 15 min, 55% B 5 min, 55–90% B 2 min, 90% B 3 min and return to initial conditions for re-equilibration (10% B 10 min). The total duration of the method was 65 min. The injection volume was 10 and 95 µL for sestonic extracts and direct injection of water samples, respectively.

Analyses were carried out in the ESI positive ionization mode. Nitrogen was used as sheath, auxiliary and collision gas. The operating conditions for LC–HRMS were previously optimized and the reproducibility of fragmentation was checked [47]. The source parameters and voltages used were: a capillary temperature of 250 °C, heater temperature of 30 °C, sheath gas flow rate of 42 psi, auxiliary gas flow rate of 10 (arbitrary units) and sweep gas flow rate of 0 (arbitrary units), spray voltage of 4.25 kV, capillary voltage of 35 V, tube lens voltage of 186 V and skimmer voltage of 35 V. Data was acquired simultaneously in the full scan and AIF modes (at 30 and 70 eV). The mass range was *m*/*z* 400–1200 in the full scan and *m*/*z* 60–1200 in the AIF mode. In the optimization of the fragmentation, all collision energies were tested, obtaining spectra similar to either the high energy (70 eV) or the medium energy (30 eV). Finally, the two collision energies that provided the most characteristic spectra were chosen. Besides, cyclic structures such as MCs, APs and NOD require high energy for their fragmentation. While, as is known, in the particular case of MCs such as MC-RR that present discharged species ([M+2H]^2+^), they fragment at medium energy. The automatic gain control (AGC) was set as “balanced” (1 × 10^6^ counts) with a maximum injection time of 250 ms. High resolution defined as R: 50,000 (*m*/*z* 200, 2 Hz, FWHM) was set in all scan events and mass accuracy expressed as parts per million (ppm) were used. Mass accuracy in all mass range *(m*/*z* 60–1200) was <5 ppm. Therefore, a maximum of ±5 ppm extraction window was allowed for peak identification. The instrument was daily externally calibrated with commercial Positive Ion Calibration solution (Thermo Fisher Scientific, Bremen, Germany). Data was processed with Xcalibur 2.1 and Trace Finder EFS 3.3 software (Thermo Fisher Scientific, Bremen, Germany).

### 4.5. Identification and Quantitation

The identification of analytes was performed according to their experimental exact mass (*m*/*z*), accurate isotopic pattern, evidence from the fragmentation data and the retention time. The chromatographic peak must be Gaussian and must have at least 10 points/peak, with a S/N >3 and area ≥10,000 count. The ratio of the chromatographic retention time of the analyte to that of the internal standard (NOD) must match those from standards of the calibration curve within ±2.5% tolerance.

The combination of high resolution and restrictive criteria was crucial for the unequivocal identification of target and unknown compounds. To ensure the reliability of the identifications, convenient HRMS and accuracy, described in Section 4.4, were employed in addition to, the following criteria: elements considered were restricted in accordance with MCs, NODs, CYN and CPs molecular formulae (C:10–60, H:15–90, O:0–20, N:0–20, S:0–1); the experimental isotopic pattern was matched regarding the theoretical in silico isotopic pattern (IP ≥ 70%); and, the charge, the RDBE and nitrogen rule were taken into account. A homemade database of 157 MCs, 10 NODs, CYN and 29 CPs (10 APs, 8 oscillapeptins, 4 microviridins, 2 agardhipeptins, 2 oscillapeptilides, OCA Y, oscillaginin and oscillacyclin) was used for identification purposes. Ion species +H, +NH_4_, +Na; +K and +2H were considered. The database included diagnostic fragments of each bioactive substance from the literature. The positive identification of the suspect analytes was confirmed by the detection of diagnostic fragments in the AIF spectra and XIC chromatograms such as *m*/*z* 135.0804 (characteristic Adda fragment) for MCs and *m*/*z* 263.1390 or 201.0982 for APs. The XIC signals of diagnostic fragments (*m*/*z*) in the AIF acquisition mode were used for quick and easy detection of signals that can be identified as cyanotoxins or/and cyanopeptides.

For quantitation purposes, the XIC signals of analytes (*m*/*z*) in the full scan acquisition mode were employed. Calibration curves and sample quantitation were based on area ratios of native compounds to NOD as IS. Due to the unavailability of CPs standards, its concentration has been estimated with respect to MC-LR (as MC-LR equivalents). The method was previously validated [47]. The limit of detection (LOD) and limit of quantitation (LOQ) were calculated individually for target MC, namely the MCs for which an analytical standard is available, includes in Table 1 and Table 2 (MC-dmRR, RR, dmLR, YR, LR WR, LA, LY, LW and LF). The limit of detection (LOD) was estimated for a signal-to-noise (S/N) ratio equal to 3 from the chromatograms of the samples spiked at the lowest validated level and the lowest level of linearity range was considered the limit of quantitation (LOQ). The LODs and LOQs of methodologies for the analysis of all target MCs in extracellular (direct injection and LC–HRMS analysis) and intracellular (preconcentration by methanol extraction and LC–HRMS analysis) fractions were 0.2 and 10 ng/L and 0.05 and 0.3 µg/L, respectively.

## Figures and Tables

**Figure 1 toxins-12-00541-f001:**
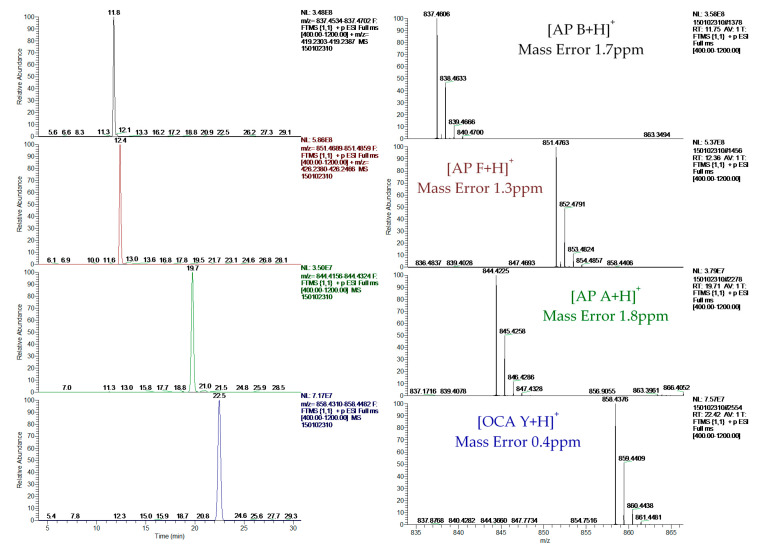
Extracted ion chromatograms and mass spectra of ocillamide Y and anabaenopeptins A, B and F detected in sample M1 (superficial scums) by LC–HRMS.

**Figure 2 toxins-12-00541-f002:**
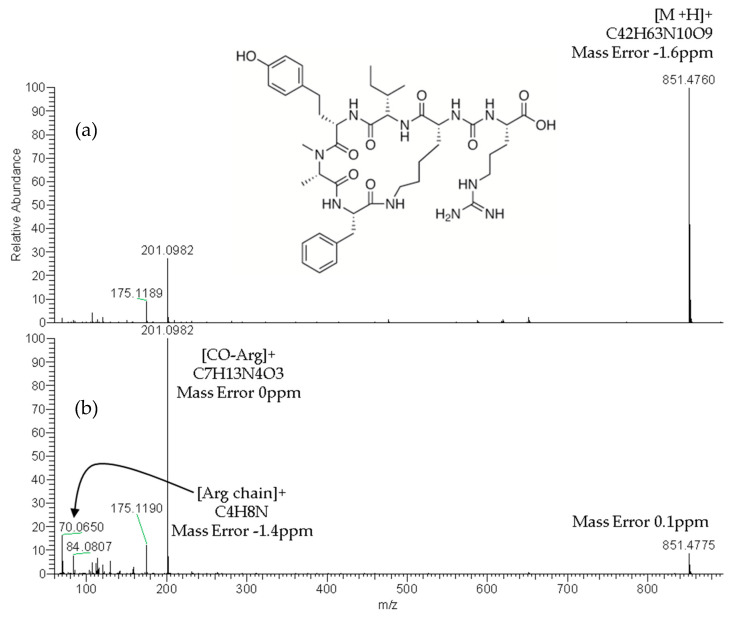
Product ion spectra of anabaenopeptin F (exact *m*/*z* of [M+H]^+^: 851.4774) detected in sample M1 (superficial scums) by LC–HRMS in the all ion fragmentation (AIF) acquisition mode at: 30 eV (**a**) and 70 eV (**b**).

**Figure 3 toxins-12-00541-f003:**
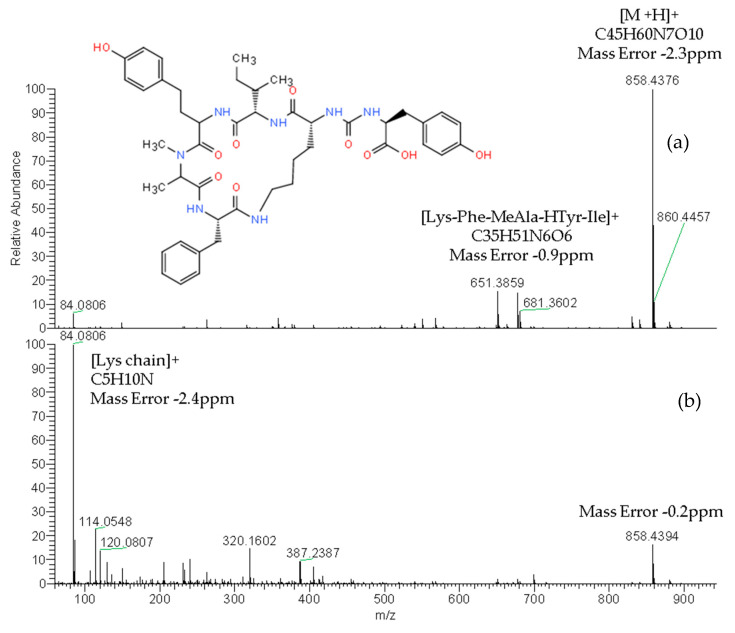
Product ion spectra of ocillamide Y (exact *m*/*z* of [M+H]^+^: 858.4396) detected in sample M1 (superficial scums) by LC–HRMS in the AIF acquisition mode at: 30 eV (**a**) and 70 eV (**b**).

**Figure 4 toxins-12-00541-f004:**
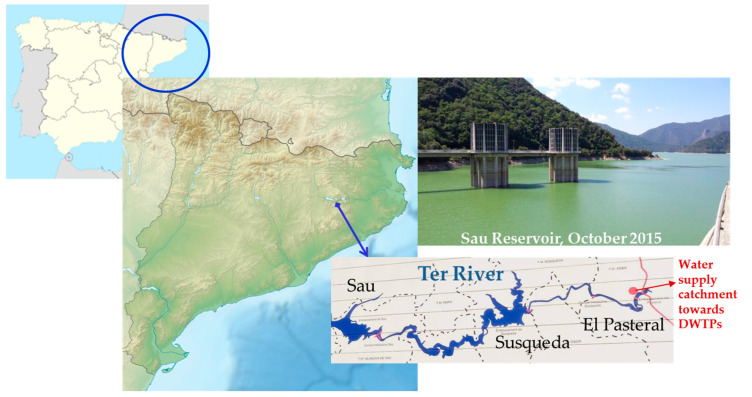
Sampling map.

**Table 1 toxins-12-00541-t001:** Results of the target analysis of microcystins obtained by LC–HRMS for aqueous fractions.

Concentration (µg/L)	Sample Reference							LOQ (µg/L)
	Input DWTP	Costa Brava Tank	Dam	A1 Water Supply Catchment	M1 Superficial Scums	M2 Reservoir Center	M3 11-m Depth	
Reservoir	El Pasteral		Susqueda	Sau				
MC-dmRR	n.d.	n.d.	n.d.	n.d.	n.d.	n.d.	n.d.	0.3
MC-RR	n.d.	n.d.	n.d.	<0.01 *	<0.01 *	<0.01 *	<0.01 *	
MC-dmLR	n.d.	n.d.	n.d.	n.d.	n.d.	n.d.	n.d.	
MC-YR	n.d.	n.d.	n.d.	n.d.	n.d.	n.d.	n.d.	
MC-LR	n.d.	n.d.	n.d.	n.d.	n.d.	n.d.	<0.01 *	
MC-WR	n.d.	n.d.	n.d.	n.d.	n.d.	n.d.	n.d.	
MC-LA	n.d.	n.d.	n.d.	n.d.	n.d.	n.d.	n.d.	
MC-LY	n.d.	n.d.	n.d.	n.d.	n.d.	n.d.	n.d.	
MC-LW	n.d.	n.d.	n.d.	n.d.	n.d.	n.d.	n.d.	
MC-LF	n.d.	n.d.	n.d.	n.d.	n.d.	n.d.	n.d.	

n.d.: not detected; MC: microcystin; LOQ: limit of quantitation (direct injection and LC–HRMS analysis); DWTP: Drinking Water Treatment Plant; * the concentration was estimated by extrapolation.

**Table 2 toxins-12-00541-t002:** Results of the target analysis of microcystins obtained by LC–HRMS for sestonic fractions of Sau samples.

Concentration (ng/L)	Sample Reference			LOQ (ng/L)
	M1 Superficial Scums	M2 Reservoir Center	M3 11-m Depth	
MC-dmRR	n.d.	n.d.	<LOQ (0.4) *	10
MC-RR	n.d.	n.d.	<LOQ (0.4) *	
MC-dmLR	n.d.	n.d.	n.d.	
MC-YR	n.d.	n.d.	n.d.	
MC-LR	n.d.	n.d.	n.d.	
MC-WR	n.d.	n.d.	n.d.	
MC-LA	n.d.	n.d.	n.d.	
MC-LY	n.d.	n.d.	n.d.	
MC-LW	n.d.	n.d.	n.d.	
MC-LF	n.d.	n.d.	n.d.	

n.d.: not detected; MC: microcystin; LOQ: limit of quantitation (pre-concentration by methanol extraction and LC–HRMS analysis); * the concentration was estimated by extrapolation.

**Table 3 toxins-12-00541-t003:** Results of the suspect screening analysis of cyanobacterial peptides obtained by LC–HRMS for sestonic fractions of Sau samples.

Estimated Concentration	Sample Reference		
	M1 Superficial Scums	M2 Reservoir Center	M3 11-m Depth µg/L
Anabaenopeptin C	2.6 **µg/L**	0.03 **µg/L**	n.d.
	**mg/L**	**mg/L**	
Anabaenopeptin B	30	0.5	0.7
Anabaenopeptin F	58	1.1	2.2
Anabaenopeptin A	4.9	0.1	0.1
Oscillamide Y	13	0.2	0.2

n.d.: not detected.

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
