# Peer review of "High Levels of Anabaenopeptins Detected in a Cyanobacteria Bloom from N.E. Spanish Sau-Susqueda-El Pasteral Reservoirs System by LC–HRMS"

_toxins, 2020, doi:10.3390/toxins12090541_

Round 1
Reviewer 1 Report
This paper is well organized and includes interesting findings. Particularly, analytical part including mass spectrometric approach should be highly evaluated. However, there are several comments and questions for better understanding as follows:
Line 49-51
It is widely accepted that anabaenopeptins and cyanopeptolins are substantially non-toxic.
Line 86-87
The following reference should be added for anabaenopeptins: Rapid Commun. Mass Spectrom. 2007; 21: 1025–1033.
Line 107-108
It is difficult to understand, particularly retrospective analysis?
Line 204
Target analysis, in this case what are targets?
Tables 1 and 2
Are these tables required?
Line 260
What is suspect screening?
Line 268
Is the home-made database available for other people?
Line 310-314
As you know, there are at least 6 components of ABPNs. Is it possible to differentiate these components plus OCA Y using the characteristic fragment ions shown here? In the fragmentation of ABPNs by MS/MS, the cleavage at ureido bond is predominant to provide a characteristic cyclic ion such as m/z 651 in the case of OCA Y. In the present case such ion was detected in OCY A, but not appeared in anabaenopeptin F. It is desirable to obtain a reproduced fragmentation. Were the operating conditions optimized?
Line 316
Ile → allo-Ile
Line 317-322
Is it possible to change to milder fragmentation to obtain diagnostic fragment ions by HCD?
Line 340 and 343
Is “fragmentation spectra” appropriate? “Production ion spectra” is better.
Line 453
peak chromatographic → chromatographic peak
Author Response
We appreciate all suggestions of reviewers. You can find below the answers and actions that have been done, concerning comments of the two reviewer 1. In bold we have copied the comments coming from the reviewer and from each comment our response is given.
REVIEWER 1:
- Line 49-51. It is widely accepted that anabaenopeptins and cyanopeptolins are substantially non-toxic.
The sentence has been changed according to reviewer suggestion:
For example, acute and chronic liver damage is associated with exposure to hepatotoxic peptides microcystins (MCs) and the alkaloid cylindrospermopsin.
As mentioned later in the line 57: Anabaenopeptins are not included in any regulations or recommendations and they are considered less toxic than MCs, which are a potent and regulated toxic, but current data show that they can give a positive response in some toxicity tests.
- Line 86-87. The following reference should be added for anabaenopeptins: Rapid Commun. Mass Spectrom. 2007; 21: 1025–1033.
Done.
The publication has been incorporate in the manuscript as reference 34.
[34] Mayumi, T.; Kato, H.; Kawasaki, Y.; Harada, K. Formation of diagnostic product ions from cyanobacterial cyclic peptides by the two-bond fission mechanism using ion trap liquid chromatography/multi-stage mass spectrometry. Rapid Commun. Mass Spectrom. 2007, 21, 1025–1033.
- Line 107-108. It is difficult to understand, particularly retrospective analysis?
The retrospective analysis was described in line 213-219 of revised manuscript. LC-HRMS operate at high resolution and full scan mode with high sensitivity which allows their capabilities in the detection of unknowns. The full scan acquisition mode enables to detect any compound that ionizes in the m/z working range, simply extracting its exact mass. With regard to environmental monitoring programs, a major advantage of LC-HRMS is the possibility of retrospective analysis of full-scan data, which enables laboratories to search for “new” contaminants years after data recording, without having to re-analyze samples.
In order to clarify the concept of retrospective analysis, this description has been added to the manuscript (line 213-219).
- Line 204. Target analysis, in this case what are targets?
MC target analysis are the determination of 10 MCs more commonly detected in blooms and those that we have a certified analytical standard (MC-dmRR, RR, dmLR, YR, LR WR, LA, LY, LW and LF). This target analysis is described in lines 142-144 and the list of target analytes is showed in Table 1 and 2. Finally, a description about the different approaches possible by HRMS (target analysis, suspect screening and non-target screening) has been added in the lines 137-140.
- Tables 1 and 2. Are these tables required?
This comment should refer to none of the target MCs have been detected at significant level of concentration. But, if possible, we would prefer to keep the tables 1 and 2 in the manuscript to graphically display the list of MCs included in the target analysis, the sample references, the quantification limits of the method and the results of target analysis by LC-HRMS in aqueous and sestonic fractions, even though most results were not detected (n.d.).
- Line 260. What is suspect screening?
LC-HRMS with high mass accuracy has emerged as a powerful tool. Highly resolved and accurate orbitrap technology allows for a more reliable target analysis with reference standards, a screening for suspected analytes without reference standards, or a screening for unknowns. In some fields, reference standards are currently not available for a large number of potential environmental contaminants, in particular transformation products. However, compound-specific information for suspects is available, such as molecular formula and structure, which can be efficiently used in the identification and confirmation process. Suspect screening is done based on a suspect ion list or database which summarizes all this information from the literature and from in silico approaches. In the present work, a home-made database of 157 MCs, 10 NODs, CYN and 29 CPs (10 ABPNAPs, 8 oscillapeptins, 4 microviridins, 2 agardhipeptins, 2 oscillapeptilides, OCA Y, oscillaginin and oscillacyclin) was used for suspect screening. Intelligent strategies allow for combining target analysis and suspects and non-target screening into the same analytical run, including the recording of product ion spectra for target and suspected compounds and intense unknown peaks by data-dependent MS/MS analysis.
Section 2.4, now section 2.2 in revised manuscript, has been modified to better describe what the suspect screening is.
- Line 268. Is the home-made database available for other people?
The home-made database includes different variants described in the literature. This database not yet published it is dynamic, and we are updating and reviewing it. Because of that, we are looking for an effective way to publish it that allows us to review and publish this database efficiently in order to the latest version with the new substances that may emerge can be easily reviewed and publicly accessible.
On the other hand, specifically, relevant references with comprehensive lists of microcystins are included in the manuscript.
Bortoli, S.; Volmer, D.A. Account: Characterization and identification of microcystins by mass spectrometry. Eur. J. Mass Spectrom. 2014, 20(1), 1–19.
Handbook on Cyanobacterial Monitoring and Cyanotoxin Analysis. Edited by: Meriluoto J, Spoof L, Codd GA. Wiley. ISBN 978-1-119-06868-6.
Bogialli, S.; Bortolini, C.; Di Gangi, I.M.; Di Gregorio, F.N.; Lucentini, L.; Favaro, G.; Pastore, P. Liquid chromatography-high resolution mass spectrometric methods for the surveillance monitoring of cyanotoxins in freshwaters. Talanta 2017, 170, 322–330.
Natumi, R.; Janssen, E.M.-L. Cyanopeptide Co-Production Dynamics beyond Mirocystins and Effects of Growth Stages and Nutrient Availability. Environ. Sci. & Technol. 2020, 54, 6063-6072.
- Line 310-314. As you know, there are at least 6 components of ABPNs. Is it possible to differentiate these components plus OCA Y using the characteristic fragment ions shown here? In the fragmentation of ABPNs by MS/MS, the cleavage at ureido bond is predominant to provide a characteristic cyclic ion such as m/z 651 in the case of OCA Y. In the present case such ion was detected in OCY A, but not appeared in anabaenopeptin F. It is desirable to obtain a reproduced fragmentation. Were the operating conditions optimized?
The strategy is to use diagnostic fragments to characterize the family of compounds that are detected, whether it is a microcystin or an anabaenopeptin or oscillamide. Therefore, these fragments are usually common to the different congeners of the same family. Then as the LC-HRMS method operate in full scan acquisition mode at two collision energies, we have all the fragment ions in the two product ion spectra to be able to identify the specific congener. For it, the two collision energies that provided the most characteristic spectra were chosen. The operating conditions for LC-HRMS were optimized and the reproducibility of fragmentation was checked. In fact, the method was previously validated (J. Chromatogr. A 2015, 1407, 76–89) and this reference has been included in line 499 of the revised manuscript.
It should be noted that as described in line 272-274 of the revised manuscript:
The fragmentation by Exactive mass spectrometer occurs in a special collision cell by high collision dissociation (HCD) without precursor mass selection (AIF mode)
Therefore, as are explained in this section, some differences with the product ion spectra by collision-induced dissociation (CID) in quadrupole collision cell have sometimes been observed.
Section 2.4 and 4.4, now Section 2.2 and 4.4 in the revised manuscript, have been modified to include this information.
- Line 316. Ile → allo-Ile
Done.
- Line 317-322. Is it possible to change to milder fragmentation to obtain diagnostic fragment ions by HCD?
In the optimization of the fragmentation, all collision energies were tested, obtaining spectra similar to either the high energy (70eV) or the medium energy (30eV). The two collision energies that provided the most characteristic spectra were chosen, 30 and 70 eV. On the other hand, cyclic structures such as microcystins, anabaenopetins and nodularins require high energy for their fragmentation. While, as is known, in the particular case of microcystins such as MC-RR that present dicharged species ([M+2H]2+), they fragment at medium energy.
Section 4.4 has been modified to include this information.
- Line 340 and 343. Is “fragmentation spectra” appropriate? “Product ion spectra” is better.
We agree with the reviewer that it is more appropriate to use “Product ion spectra”. Therefore, at the suggestion of the reviewer we have changed “fragmentation spectra” for “Product ion spectra” in the revised manuscript.
- Line 453. peak chromatographic → chromatographic peak
Done.

Reviewer 2 Report
The manuscript describes the analysis of cyanotoxins by targeted and suspect screening methods using high-resolution mass spectrometry in the context of an urban reservoir management plan. The presented method and enables the quantification of multiple microcystins and the detection of other families of cyanopeptides produced by toxic cyanobacteria. Very interesting results were obtained towards the detection of high concentrations of anabaenopeptins where microcystins concentrations were negligible and toxicity assessment the significant contribution of anabaenopeptins toxicity. Overall, this paper is an important contribution in the field of cyanotoxins assessment by including and detecting uncommon cyanopeptides aside of microcystins showing the importance of expanding the study of cyanotoxins as part of a monitoring plans. However, some points should be mentioned on the presentation given that validation data is missing and the result and discussion section needs to be revised. Please refer to the following comments:
- Abstract and Introduction: The authors included a suspect screening study including not only anabaenopeptins but other cyanopeptides families. This experiment should be pointed out in the abstract and introduction as this part of this study is quite important in the presented results and for its originality.
- Results and discussion: The order of the information in this section is inadequate. The results should be presented first and then discussed. The paragraph in lines 120-130 and section 2.1 are more of a discussion about the presence and evolution of the bloom and should be put after the presentation of the results. Section 2.2 is more of an explanation of the management plan leading to sample analysis. This section is between the Method section and Discussion section. Please revise this section to incorporate in the discussion after results presentation and/or Method section. Overall, these paragraphs (lines 120-203) are quite heavy in the beginning of text before showing the results and should be revised for a better fluidity and presentation of the results and its discussion.
- Line 19-117… : Cyanotoxins are not metabolites properly speaking. At some extent, we can call them secondary metabolites, cyanotoxins for the most common compounds with known toxicity or cyanopeptides for the polypeptides compounds with known and unknown toxicity. Please revise the term throughout the manuscript.
- Line 16-110-409…: MC-dmLR and MC-dmRR should be written [Asp3]MC-LR and [Asp3]MC-RR, or [Dhb7]MC-LR and [Dhb7]MC-RR depending of the site of demethylation. Please, specify which compound they are and make the change throughout the manuscript.
- Line 38: Please modify crescent for another term (growing or increasing).
- Line 50: Anabaenopepin acronym is usually AP according to literature.
- m/z should be in italic.
- Line 109: The proper term when referring to the choice of resolution when using HRMS is resolving power.
- Line 216 and 477: Please specify how the LOD and LOQ were estimated in the method section and is it only for MC-LR? In line 477, LOD is mentioned but not LOQ, please add in the text. Not only MC-LR was analyzed as other microcystins and NOD were also individually analyzed. Their LODs and LOQs should be individually included in the manuscript.
- LC-HRMS method: Is the quantitative method a new method developed for this study or the method was previously validated and published? If so, please refer to the method validation, and if not, some validation information would be required: recovery from extraction, matrix effects, accuracy, repeatability and reproducibility, calibration linearity.
Author Response
We appreciate all suggestions of reviewers. You can find below the answers and actions that have been done, concerning comments of the two reviewer 2. In bold we have copied the comments coming from the reviewer and from each comment our response is given.
REVIEWER 2:
- Abstract and Introduction: The authors included a suspect screening study including not only anabaenopeptins but other cyanopeptides families. This experiment should be pointed out in the abstract and introduction as this part of this study is quite important in the presented results and for its originality.
As suggested by the reviewer, abstract and introduction have been modified to describe that a suspect screening study including not only anabaenopeptins but other cyanopeptides families were performed.
- Results and discussion: The order of the information in this section is inadequate. The results should be presented first and then discussed. The paragraph in lines 120-130 and section 2.1 are more of a discussion about the presence and evolution of the bloom and should be put after the presentation of the results. Section 2.2 is more of an explanation of the management plan leading to sample analysis. This section is between the Method section and Discussion section. Please revise this section to incorporate in the discussion after results presentation and/or Method section. Overall, these paragraphs (lines 120-203) are quite heavy in the beginning of text before showing the results and should be revised for a better fluidity and presentation of the results and its discussion.
As suggested by the reviewer, overall, the paragraphs of results and discussion have been reorganized and modified for a better fluidity and presentation of the results and its discussion.
The order of the information in this section have been changed. Firstly, the results have been explained (target and suspect screening analysis by LC-HRMS are now sections 2.1 and 2.2 in the revised manuscript, respectively). And then, the discussion about the presence and evolution of the bloom (paragraph in lines 126-132 and section 2.1) and the assessment and management tasks (section 2.2) have been placed after the presentation of the results, becoming the sections 2.3 and 2.4 in the revised manuscript, respectively. Finally, the section of assessment and management tasks has been simplified.
- Line 19-117: Cyanotoxins are not metabolites properly speaking. At some extent, we can call them secondary metabolites, cyanotoxins for the most common compounds with known toxicity or cyanopeptides for the polypeptides compounds with known and unknown toxicity. Please revise the term throughout the manuscript.
According to the reviewer's suggestion, the terms metabolite, cyanotoxins, cyanobacterial peptides and cyanopeptides have been revised throughout the manuscript. Some changes have been made, especially for the term metabolite (marked with the track changes mode of the Word in revised manuscript).
- Line 16-110-409: MC-dmLR and MC-dmRR should be written [Asp3]MC-LR and [Asp3]MC-RR, or [Dhb7]MC-LR and [Dhb7]MC-RR depending of the site of demethylation. Please, specify which compound they are and make the change throughout the manuscript.
We agree with the reviewer, but regret that we cannot specify the used microcystin variant as the certificate of analysis for these standards does not provide this data. The certificates described the microcystins (MC-dmLR and MC-dmRR) just as demethylated (dm) variants, without specifying the position of the loss of the methyl group.
- Line 38: Please modify crescent for another term (growing or increasing).
As suggested by the reviewer, the term has been replaced by growing.
- Line 50: Anabaenopeptin acronym is usually AP according to literature.
Done. But, it is worth mentioning that anabaenopeptins have been presented with different acronyms in the literature (AP, Apt, ABPN). For example, ABPN acronym was used in Harmful Algae 83 (2019) 42–94. Standardization and harmonization would be necessary to avoid it.
- m/z should be in italic.
Done. We fully agree.
- Line 109: The proper term when referring to the choice of resolution when using HRMS is resolving power.
Regarding resolution and resolving power terms, we have followed the recommendation of the reviewer, although there is controversy in the literature for this issue, because the standard nomenclature for definitions of resolution and resolving power have been interchanged over the years.
- Line 216 and 477: Please specify how the LOD and LOQ were estimated in the method section and is it only for MC-LR? In line 477, LOD is mentioned but not LOQ, please add in the text. Not only MC-LR was analyzed as other microcystins and NOD were also individually analyzed. Their LODs and LOQs should be individually included in the manuscript.
The LOD and LOQs were calculated individually for target microcystin, namely the microcystins for which an analytical standard is available, includes in Table 1 and 2 (MC-dmRR, RR, dmLR, YR, LR WR, LA, LY, LW and LF). The limit of detection (LOD) was estimated for a signal-to-noise (S/N) ratio equal to 3 from the chromatograms of the samples spiked at the lowest validated level and the lowest level of linearity range was considered the limit of quantitation (LOQ).
This information has been included in the manuscript (lines 500-507, method Section 4.4).
- LC-HRMS method: Is the quantitative method a new method developed for this study or the method was previously validated and published? If so, please refer to the method validation, and if not, some validation information would be required: recovery from extraction, matrix effects, accuracy, repeatability and reproducibility, calibration linearity.
The method was previously validated (J. Chromatogr. A 2015, 1407, 76–89). The method validation has been referenced in line 499.

Round 2
Reviewer 2 Report
The manuscript has been appropriately edited based on the first revision.